# Mixed-methods evaluation of a behavioural intervention package to identify and amend incorrect penicillin allergy records in UK general practice

Marta Wanat ,[1] Marta Santillo ,[1] Ushma Galal,[1] Mina Davoudianfar,[1] Emily Bongard ,[1] Sinisa Savic ,[2] Louise Savic,[2] Catherine Porter,[3] Joanne Fielding,[3] Christopher C Butler ,[1] Sue Pavitt,[4] Jonathan Sandoe,[3] Sarah Tonkin-Crine ,[1,5] On behalf of the ALABAMA team

MW and MS are joint first authors.
JS and ST-C are joint senior authors.

For numbered affiliations see end of article.

**Correspondence to**
Dr Marta Wanat;
marta.wanat@phc.ox.ac.uk

## ABSTRACT

**Objectives** About 6% of the UK general practice population has a record of a penicillin allergy but fewer than 10% of these are likely to be truly allergic. In the ALABAMA (Allergy Antibiotics and Microbial resistance) feasibility trial, primary care patients with penicillin allergy were randomised to penicillin allergy assessment pathway or usual care to assess the effect on health outcomes. A behavioural intervention package was developed to aid delabelling. This study aimed to investigate patients' and clinicians' views of penicillin allergy testing (PAT).

**Design** We conducted a mixed-methods process evaluation embedded within the ALABAMA trial, which included a clinician survey, a patient survey (at baseline and follow-up) and semistructured interviews with patients and clinicians.

**Settings** The study was conducted in primary care, as part of the feasibility stage of the ALABAMA trial.

**Participants** Patients and primary care clinicians.

**Results** Clinicians (N=53; 52.2%) were positive about PAT and its potential value but did not have previous experience of referring patients for a PAT and were unsure whether patients would take penicillin after a negative allergy test. Patients (N=36; 46%) were unsure whether they were severely allergic to penicillin and did not fear a severe allergic reaction to penicillin. Clinician interviews showed that they were already aware of the benefit of PAT. Interviews with patients suggested the importance of safety as patients valued having numerous opportunities to address their concerns about safety of the test.

**Conclusions** This study highlights the positive effects of the ALABAMA behavioural intervention for both patients and clinicians.

**Trial registration number** NCT04108637; ISRCTN20579216; Pre-results.

## INTRODUCTION

A total of 6%–10% of primary care patients in the UK have a penicillin allergy record[1 2] but fewer than 10% of these patients are likely to be allergic when formally tested,[3] which means that a large proportion of these people

## STRENGTHS AND LIMITATIONS OF THIS STUDY

⇒ The study benefits from using both survey and interviews, providing complementary perspectives.
⇒ The response rate for both clinician (52.2%) and patient surveys (46%) was relatively low.
⇒ The study would have benefited from a follow-up survey with clinicians, which was not possible due to study delays.
⇒ The study recruited from one region in England and includes small numbers due to issues with recruitment and testing availability; hence the results should be extrapolated with caution.
⇒ Not all trial participants completed the survey, meaning that only views of participants who chose to complete the survey are represented; additionally, both patient and clinician interview participants were a convenience sample recruited within a feasibility study, hence not allowing to capture perspectives of individuals who did not want to take part in the interviews.

are unnecessarily avoiding first-line antibiotic treatments. Penicillin allergy testing (PAT), usually undertaken in specialist immunology clinics, offers the opportunity to confirm or discount a penicillin allergy, although such services are only currently available for a limited number of patients. The negative consequences of having a penicillin allergy record include: suboptimal clinical outcomes, increased antibiotic resistance[4–6] and higher healthcare costs.[5 7 8]

Barriers to removing incorrect penicillin allergy records include lack of standardised protocols across practices,[9 10] lack of access to skin testing (ST)[11] and the difficulty of convincing patients that the record is no longer needed after a negative test.[12 13] These studies also suggest multiple ways to improve research, testing and implementation of

BMJ

penicillin allergy delabelling initiatives such as easier access to ST reagents, standardised ST protocols and a greater understanding of penicillin allergy epidemiology.

There is limited understanding of the views of patients and primary care healthcare professionals on delabelling initiatives in primary care. Specifically, it would be helpful to have a better understanding of their views about simplifying and standardising ST protocols for use only in high risk patients and performing routine direct oral challenges in individuals with low-risk histories to confirm current tolerance. It is necessary to develop a greater appreciation in patients and clinicians of the well-established epidemiology of suspected penicillin allergy, particularly how rarely it is confirmed. The National Institute for Health and Care Research (NIHR)-funded Allergy Antibiotics and Microbial resistance (ALABAMA) programme includes a randomised controlled trial of primary care patients who are randomised to a pre-emptive penicillin allergy assessment pathway (PAAP) or usual clinical care to assess the effect on health outcomes.

Within this study, we addressed the current gap, by investigating the perspective of both patients and clinicians on PAT as part of the mixed-methods process evaluation[14 15] of the ALABAMA feasibility trial. Our mixed-methods process evaluation included a clinician questionnaire, a patient questionnaire and interviews with clinicians and patients.

## METHODS

The ALABAMA feasibility trial involved 11 general practices based in the Leeds/Bradford area of England. The behavioural intervention package was designed to target both general practice clinicians responsible for the prescription of antibiotics and patients with a suspected incorrect penicillin allergy record, and was an integral part of the PAAP. The details on the process, methods and outcomes of the development of the behavioural package of the ALABAMA intervention has been published elsewhere.[16] The PAAP targets patients assessed as 'low risk' of having a true penicillin allergy, that is with no history of anaphylaxis or other severe allergic reactions. It also aimed to streamline the test process by undertaking patient history screening in general practice (stage 1) and introducing an efficient one-stop procedure at a hospital immunology clinic for PAT. The PAT included either an ST (stage 2) and oral challenge test (OCT) involving taking oral doses of a penicillin solution while supervised (stage 3), or proceeding directly to an OCT, depending on the individual patient history (the majority of patients). Following the PAT, patients and practices received confirmation of a patient's allergy status by letter from the immunology clinic.

### Intervention

Patients in the intervention arm (PAAP arm) received a booklet about allergy testing (pretest booklet), completed the two/three stages of PAAP (screening/ patient history, skin test and/or OCT) and, if tested negative, received a booklet and a card giving information about their result. The pretest booklet informed patients about incorrect allergy records, how they may benefit from having a PAT and what was involved. The post-test booklet informed on the reliability of test results and consequences of a negative test result. A laminated card was provided stating the negative allergy result which could be shown to General Practicioners (GPs) and other prescribers.

Clinicians' booklet aimed to increase their motivation to refer patients to PAT and prescribe penicillin after a negative PAT result, together with a letter to inform them about the patient test results and how to change allergy records when the test result was negative. A computer pop-up alerting a GP to the change in the patient's allergy status was built into the electronic patient record system (SystmOne, TPP).

### Patient and public involvement

The ALABAMA programme of work has a patient advisory group advising the research team on all aspects of the study.

### Mixed-methods evaluation using surveys and interviews

#### Clinician survey

Twelve-item surveys on a 7-point Likert scale (online supplemental appendix 1) were sent to all clinician participants in the 11 practices. The survey was not validated but its items were informed by constructs from the theoretical domains framework[17] and measured clinicians' knowledge, cognitive skills, intentions, beliefs about consequences, professional role and identity, goals and social influences related to PAT. Clinicians were invited to take part in the survey by email at two time points; at the start of the feasibility trial and once all patient participants from a practice had been followed up for 4 months postrandomisation. Clinicians completed the survey online (Survey Monkey).

#### Patient survey

Surveys were sent to all patient participants in the 11 practices. The patient survey included 13 items on a 7-point Likert scale, where 7 was strongly agree and 1, strongly disagree (online supplemental appendix 2). Patients were asked about their beliefs concerning their penicillin allergy status, their views on possible reactions during a PAT, and their intentions to take penicillin in the future in the case of a negative test result.

The aim of the surveys was to compare participants' knowledge and beliefs about penicillin allergy and PAT before and after the feasibility trial. Patients completed the questionnaire at two time points. Patients completed the belief questionnaire at baseline over the telephone with a member of the trial team. They completed the follow-up questionnaire 28–30 days postrandomisation over the telephone with a member of the trial team.

## Analysis

Patient and clinician data were extracted and analysed using SPSS (V.26). Medians for each group and items were reported. A series of non-parametric tests (Wilcoxon signed rank test) were conducted to compare whether the medians for each item when compared with the midpoint of the scale (one sample test), and whether the medians for each item differed between groups (independent samples) and at different time points (related samples).

## Interviews

We conducted semistructured interviews using two topic guides (one for patients, one for clinicians; online supplemental appendix 3), which were informed by the literature on penicillin allergy.[15] After obtaining written consent, interviews were conducted over the telephone by two experienced female qualitative researchers (MW and MS) wo were not part of the trial. The interviews were audiorecorded and transcribed verbatim. Interviews continued until data indicated saturation.[18]

## Clinicians

One clinician from each of the 11 practices was invited to take part in an interview. This was the study champion, who led on the patient recruitment and was often responsible for updating patients' records. Clinicians were invited by email. Clinicians were asked about their views and experience of referring patients to PAT, prescribing penicillin after a negative test result and their experience in taking part in the trial and intervention materials.

## Patients

Patients were invited to take part in an interview by letter or telephone. We sampled patients in the intervention arm, to include people with positive and negative test results. Patients were asked their views and experiences of undergoing PAT, their experience of taking part in the trial and their views on intervention materials.

## Interview analysis

Transcripts were analysed using inductive thematic analysis.[19] Transcripts were read and reread by MW and MS during and after data collection. A coding manual was developed after the first few interviews and was refined during the analytical process. To enhance the trustworthiness of data, analysis was discussed by a multidisciplinary team consisting of psychologists, a primary care clinician and colleagues from hospital-based immunology with expertise in penicillin allergy and microbiology services.

# RESULTS

## Clinician survey

Fifty-three clinicians responded to the survey (43 GPs, 9 nurse practitioners and 1 pharmacist), response rate of 52.2%. Results for each item are presented in table 1.

**Table 1** Clinician survey responses

| Questions | All respondents Median (IQR) | GPs (n=43) Median score (IQR) | Nurses (n=9) Median score (IQR) | Pharmacists (n=1) Score |
|---|---|---|---|---|
| 1. I understand what is involved in penicillin allergy testing | 5 (4–6) | 5 (4–6) | 5 (2.5–6) | 4 |
| 2. I have previous experience of referring patients for Penicillin allergy testing | 2 (1.5–5) | 2 (2–5) | 2 (1.5–4) | 1 |
| 3. I am happy to refer patients for Penicillin allergy testing | 6 (5–7) | 6 (6–7) | 6 (4–7) | 2 |
| 4. Penicillin allergy testing can benefit my patients | 6 (6–7) | 7 (6–7) | 6 (5.5–7) | 6 |
| 5. Penicillin allergy testing can benefit my practice | 6 (6–7) | 6 (6–7) | 6 (6–6.5) | 6 |
| 6. It is safe to prescribe penicillin after a negative PAT | 6 (4–6) | 6 (5–6) | 4 (4–6.5) | 4 |
| 7. I am confident in discussing PAT results with my patients | 5 (4–6) | 5 (4–6) | 4 (4–6) | 2 |
| 8. I am happy to change patient records based on the results of penicillin allergy testing | 6 (4.5–6) | 6 (5–7) | 5 (4–6) | 4 |
| 9. I would prescribe penicillin, if indicated, to patients with a negative PAT result | 6 (5–6) | 6 (5–6) | 5 (4–6.5) | 4 |
| 10. My colleagues support de-labelling of patients with incorrect penicillin allergy status using penicillin allergy testing | 5 (4–6) | 6 (4–6) | 4 (4–7) | 5 |
| 11. My patients would be happy to be referred for penicillin allergy testing | 6 (4–6) | 6 (4–6) | 5 (4–6) | 5 |
| 12. My patients would be happy to take penicillin following a negative PAT | 4 (4–6) | 4 (4–6) | 4 (4–6) | 5 |

Scale 1=strongly disagree, 2=disagree, 3=slightly disagree, 4=not sure, 5=slightly agree, 6=agree, 7=strongly agree.
GPs, General Practicioners; PAT, penicillin allergy test.

Clinicians gave positive responses to the majority of items. In general, they did not have previous experience of referring patients for a PAT, and were unsure whether patients would accept a prescription of penicillin after a negative PAT. Results suggested that GPs and nurses had slightly different beliefs. GPs were more likely than nurses to indicate that they would be happy to refer patients for a PAT, change allergy records and prescribe penicillin, following a negative test result. Nurses seemed less certain that a PAT would benefit their patients, more unsure about the safety of prescribing penicillin after a negative result, more unsure whether colleagues would support delabelling and less confident about discussing PAT results with patients.

### Patient survey

Seventy-nine patients participated in the feasibility study, of which 36 (46%) completed the baseline questionnaire. Nineteen patients were randomised to the intervention arm. Fifteen of these completed the PAT (2 positive, 13 negative). Ten of these patients completed the follow-up survey (all PAT negative). We checked whether survey responses from participants in the intervention and control arms were similar at baseline. As there were no significant differences, we then compared responses at baseline and follow-up for the 10 patients in intervention arm who answered both surveys. Five items indicated that participant beliefs had changed (table 2).

Patients were already confident about the safety and accuracy of the test at baseline, however they were unsure whether they were severely allergic to penicillin and but did not fear a severe allergic reaction to penicillin.

At follow-up patients were more likely to believe that penicillin is the best treatment for bacterial infections; that they would not have a severe allergic reaction to penicillin; and, were less scared of having a severe reaction than before they did the test. Moreover, patients were more confident that they were not allergic to penicillin. Lastly, patients were more likely to agree that the majority of patients with penicillin allergy records in the UK might not be allergic.

### Interviews with patients and clinicians

We conducted 17 interviews, 7 interviews with clinicians and 10 with patients. We invited 12 patients to an interview but two have not responded to an invitation. The interviews have been conducted between November 2019 and April 2020 and lasted between 15 and 40 min. Table 3 presents participant characteristics.

#### Clinicians
##### Theme 1: motivations to join the study – penicillin allergy is a problem in primary care

Clinicians highlighted that penicillin allergy is a 'major problem' in primary care resulting in fewer antibiotic options for patients and difficulty in prescribing, particularly for patients with multiple allergies and co-morbidities. They were aware that a number of allergy records may be incorrect but had limited access to allergy services. Hence, they saw the study as an opportunity to revoke erroneous labels which otherwise would not be possible.

> I think it's a huge problem. […] there's lots of penicillin allergy recorded on patients' notes, a lot of which I think is probably incorrect, but there's no way of proving or disproving it unless they're in this study at the moment or unless this study gets taken further. [P2, GP]

##### Theme 2: extent of confidence in allergy testing

Clinicians had confidence in the safety of allergy testing as they believed that the hospital environment was an appropriate place for testing to take place. Also, they reported that testing had components which were familiar to them. Finally, they believed the screening procedures included enough safety netting that patients at risk of a severe reaction would not get to the oral challenge stage:

> Getting to the stage of having to take a penicillin dose orally but there are steps before that to mitigate the risk. I haven't got any particular worries about it, no. [P4, GP]

##### Theme 3: experience of discussions with patients about allergy testing

Clinicians felt that the training they received as part of the study allowed them to answer patients' questions. Clinicians found the pretest booklet informative and some used it in their discussions with patients, to explain the study and potential benefits from the patients' perspective:

> Very well prepared actually, from the information provided and also with the information sheets that are given to patients […] I found that very helpful to look up and read beforehand, so the glossy leaflets about what to expect if you were going to the penicillin testing, so it was helpful to read through that so I could talk about that, from a patient's perspective. [P4, GP]

Clinician's confidence in the safety of testing meant that they also felt comfortable and confident in reassuring patients, if they asked about safety. They also reported that patients who responded to the study were very keen to get tested and find out their allergy status. They highlighted that proactively calling patients to follow up on the postal invite facilitated recruitment.

##### Theme 4: implementation of procedures in a practice – extent of staff involvement

While in each practice there was a GP lead who was the main point of contact for the study team, on a day-to-day basis it meant the champion was often the only person responsible for identifying and consenting patients. GPs highlighted the importance of keeping everyone informed about the study, especially in relation to

**Table 2** Comparing patient survey responses at baseline and follow-up for patients in the PAAP and usual clinical care arms of the trial

| Questions | Baseline (N=36) | | | | PAAP only (N=10) | | |
| --- | --- | --- | --- | --- | --- | --- | --- |
| | PAAP median (N=13) (IQR) | UCC median (N=23) (IQR) | P value* | | Baseline (n=10) Median (IQR) | Follow-up (n=10) Median (IQR) | P value† |
| 1. Penicillin is often the best treatment for bacterial infections. | 2 (1–4) | 2 (1–4) | 0.902 | | 2 (1–4) | 1 (1–2) | 0.026 |
| 2. Having an incorrect penicillin allergy label can be harmful for patients. | 2 (1–2) | 2 (1–3) | 0.214 | | 2 (1–3) | 1 (1–2) | 0.107 |
| 3. Having a penicillin allergy test is not beneficial for patients themselves | 6 (6–7) | 6 (5.5–6.5) | 0.931 | | 6 (5.5–7) | 6 (1.75–6.25) | 0.477 |
| 4. Most people in the UK who think they are allergic to penicillin are not actually allergic. | 4 (2–4) | 4 (3–4) | – | | 4 (3.5–6) | 2 (1.75–4) | 0.070 |
| 5. I believe that I am allergic to penicillin. | 4 (4–4) | 2 (1.5–5) | 1.000 | | 2 (1.5–5) | 6 (6–6.25) | 0.012 |
| 6. There is a high chance that I would have a serious allergic reaction if I took penicillin. | 4 (2–4) | 4 (3–6) | 0.722 | | 4 (3.5–6) | 6 (5–7) | 0.031 |
| 7. It is important for me to know whether or not I am allergic to penicillin. | 1 (1–1) | 1 (1–2) | 0.610 | | 1 (1–1) | 1.5 (1–2) | 0.414 |
| 8. Penicillin allergy testing is safe. | 2 (2–3) | 2 (2–2) | 0.368 | | 2 (2–2) | 1 (1–2) | 0.527 |
| 9. Penicillin allergy testing can say if someone is allergic to penicillin or not. | 2 (1–2) | 2 (1.5–2) | 0.756 | | 2 (1.5–4) | 2 (1–2) | 0.655 |
| 10. I would benefit from being able to take penicillin antibiotics safely when needed in future. | 1 (1–2) | 2 (1–2) | 0.951 | | 2 (1–2) | 1.5 (1–2) | – |
| 11. I am frightened about having a serious allergic reaction if I take penicillin. | 6 (3–6) | 5 (3–6) | 0.297 | | 5 (3.5–6) | 6 (6–6.25) | 0.026 |
| 12. My doctor would be happy to prescribe penicillin if a test showed I was not allergic to penicillin | 2 (2–2) | 2 (1.5–2) | 0.464 | | 2 (2–2) | 2 (1–2) | 0.317 |
| 13. I would be happy to take penicillin prescribed by my doctor if a test showed I was not allergic | 2 (1–2) | 2 (1–2) | – | | 2 (1–2) | 1.5 (1–2) | 0.705 |

1=strongly agree; 2=agree; 3=slightly agree; 4=not sure; 5=slightly disagree; 6=disagree; 7=strongly disagree; –= no result.
*Median test, k sample.
†Wilcoxon Signed Rank Test.
PAAP, penicillin allergy assessment pathway; UCC, Usual Clinical Care.

**Table 3** Interview participant characteristics

| | Role (clinician only) | Mean age (years) | Age range (years) | Gender (female, n (%)) | Role (clinician only) | Tested negative (patient only) |
|---|---|---|---|---|---|---|
| Patients (N=10) | | 63.7 | 30–82 | 8 (80) | N/A | 8/10 |
| Clinicians (N=7) | 6 GPs; 1 advanced nurse practitioner | 46.4 | 43–57 | 4 (57) | 6 GPs; 1 advanced nurse practitioner | N/A |

GPs, General Practicioners; N/A, not applicable.

recording a negative test result and amending allergy records to facilitate future penicillin prescribing:

> I made sure that all my colleagues knew that any letter regarding the study had to come to me, […] so I processed all the SystmOne input, made sure that everything was clear. I made a suggestion to the team that I add some additional text regarding the fact that the patient was part of the study and whether they tested positive or not. [P1, GP]

### Patients
#### Theme 1: reasons for wanting to have access to testing
The main motivation to take part in the study was to have a definite answer on whether they were allergic or not. Patients had varying knowledge about the benefits of penicillin; some were aware that it would be beneficial for them to have access to a wider range of antibiotics and described this as one motivation to get tested. Undergoing testing to help tackle antibiotic resistance was mentioned very rarely and likely prompted by the patient pretest booklet. Patients who initially had less knowledge about the benefits of penicillin reported being encouraged to take part after reading the study information:

> After reading all the information and hearing that it's one of the best antibiotics, then I just thought, 'Well, if I can actually take it safely, then it's probably a good thing to see if I can get tested or not, so I'll take part.' [P8, Negative allergy test]

#### Theme 2: considering the safety of the test
The main concern among patients was the safety of the test. However, they also highlighted that clear and detailed information about testing procedures beforehand was helpful in alleviating their concerns. They valued multiple ways of learning about the process and safety of testing: during the consent procedure with their GP, from trial materials and from the pretest booklet:

> [The Pre-test booklet] explained everything that would happen and the time scale and how it would help me and why it would help me; so yes, it affected my decision to take part [P7, Negative allergy test].

Despite having this information before attending for the test, patients still appreciated having the process explained to them by nurses and clinicians in the hospital.

Clear information presented in lay terms put patients at ease.

#### Theme 3: acceptance of the test result
Patients seemed to accept the test result, regardless of whether it was positive or negative and indicated that, where applicable, they intended to take penicillin when prescribed in the future. Some explained that they would still exercise caution and look out for any possible signs of allergy while still having confidence in the test:

> I would sort of think, 'right, I'll take it, I'll make sure that I check everything is okay, you know, am I starting to feel anything.' But because I took it for the few days afterwards and I took quite a few doses, I feel quite okay about it because I didn't get anything at all. So I'd feel quite happy about it. I wouldn't be worried [P4, Negative allergy test]

### DISCUSSION
This study provides an in-depth understanding of patients and primary care clinicians' views of the key barriers and facilitators to referring to and attending PAT and prescribing/taking penicillin after a negative test. It also highlights the positive effects of the behavioural intervention for both patients and clinicians.

Clinicians who took part in the study were motivated by having access to testing as they were convinced of the benefits of penicillin for their patients and their practice. In line with other studies, clinicians were often unsure about criteria for referral or what allergy testing involves,[20 21] and reported little prior experience and only some knew what was involved in the test prior to the trial.

Our survey results also suggested that nurses might be more tentative about referring patients and changing an allergy status on medical records. Studies with hospital nurses suggested that they could play a role in antibiotic stewardship activities, including accurate and detailed documentation of penicillin allergy.[22 23] However, more studies are needed to explore the views of this group of healthcare professionals and better define the multidisciplinary approach to delabelling.[24]

While the baseline survey showed that patients had positive beliefs about testing, including safety, these beliefs seemed to be strengthened after they had completed the PAAP. In line with other studies,[23–25] qualitative interviews also highlighted the importance of safety of testing

as patients valued having numerous opportunities to address their concerns about safety of the test, including conversations with their GP and later with hospital staff.

Our study identified a number of important clinical implications. First, we note the importance of clinician training through provision of educational materials to ensure successful implementation of referral to PAT and subsequent penicillin prescribing in primary care. The information materials provided were useful to clinicians and increased their knowledge and confidence in referring people and having discussions with patients about PAT. Second, it also raises the importance of a clear plan in relation to the delabelling process, involving clear procedures for informing other colleagues about test results to facilitate penicillin prescribing in the future. Our study indicates that communication between staff member is crucial with dedicated roles being key. Third, while patients may have a number of concerns about safety of PAT,[13 21 26] these concerns can be successfully addressed and educational materials may be useful in increasing patients' motivation; however, primary care clinicians need to be provided with appropriate resources and time to address these issues.

## Strengths and limitations

The study benefits from using both survey and interviews, providing complementary perspectives. Including qualitative studies is important in trial and implementation settings, where they are now recommended as part of the mixed-methods evaluations[14]

However, the response rate, and the number of participants in both clinician and patient surveys was low (52.2%; N=26% and 46%; N=53, respectively), thus, the results may have to be interpreted with caution, especially as both patients and clinicians taking part in the survey are already motivated to take part in the trial. Our study would have also benefited from a follow-up survey with clinicians; this was not possible due to study delays. The study recruited from one region in England and includes small numbers due to issues with recruitment and testing availability; hence the results should be extrapolated with caution. Not all trial participants completed the survey, meaning that only views of participants who chose to complete the survey are represented. Both patients and clinician interview participants were a convenience sample recruited within a feasibility study, hence now allowing to capture perspectives of patients and clinicians who did not want to take part in the interviews, which means that these findings may have to be interpreted cautiously in terms of their transferability to other settings.

## CONCLUSIONS

This study identified a number and issues of which need to be addressed to ensure successful implementation of penicillin allergy delabelling programmes. However, the study highlighted that both patients and clinicians'

concerns can be addressed with help of carefully designed intervention materials.

**Author affiliations**
[1]Nuffield Department of Primary Care Health Sciences, Oxford University, Oxford, UK
[2]University of Leeds and Leeds Teaching Hospitals NHS Trust, Faculty of Medicine and Health, Leeds, UK
[3]Healthcare Associated Infection Group, University of Leeds and Leeds Teaching Hospitals NHS Trust, Leeds, UK
[4]Dental Translational and Clinical Research Unit, Faculty of Medicine and Health, University of Leeds, Leeds, UK
[5]National Institute for Health Research (NIHR) Health Protection Research Unit (HPRU) in Healthcare Associated Infections and Antimicrobial Resistance, University of Oxford, Oxford, UK

**Collaborators** The ALABAMA team consists of: Jenny Boards, Emily Bongard, Christopher C Butler, Mina Davoudianfar, Mandy East, Joanne Fielding, Philip Howard, Sue H Pavitt, Catherine E Porter, Jonathan AT Sandoe, Marta Santillo, Louise Savic, Sinisa Savic, Bethany Shinkins, Sarah Tonkin-Crine, Marta Wanat, Robert West and Ly-Mee Yu.

**Contributors** Conceptualisation, CCB, SP, JS and ST-C; Formal analysis, MW, MS and UG; Funding acquisition, CCB, SP, JS and ST-C; Investigation, MW, MS, MD, EB, SS, LS, CP, JF, CCB, SP, JS and ST-C; Methodology, MW, MS and ST-C; Project administration, CP and JF; Resources, SP and JS; Supervision, ST-C; writing-original draft, MW and MS; writing-review and editing, UG, MD, EB, LS, CP, JF, CCB, SP, JS and ST-C. ST-C is responsible for the overall content as the guarantor.

**Funding** This work was supported by the National Institute for Health Research (NIHR) under its Programme Grants for Applied Research Programme (Grant Number RP-PG-1214-20007) and by the National Institute for Health Research (NIHR) Health Protection Research Unit in Healthcare Associated Infections and Antimicrobial Resistance (NIHR200915) at the University of Oxford in partnership with the UK Health Security Agency (UKHSA) (ST-C and CCB) and by the National Institute for Health Research (NIHR) infrastructure at Leeds.

**Disclaimer** The funders had no role in the design of the study; in the collection, analyses, or interpretation of data; in the writing of the manuscript, or in the decision to publish.

**Competing interests** None declared.

**Patient and public involvement** Patients and/or the public were involved in the design, or conduct, or reporting, or dissemination plans of this research. Refer to the Methods section for further details.

**Patient consent for publication** Not applicable.

**Ethics approval** The study received ethical approval from the London Bridge Research Ethics Committee (Ref: 19/LO/0176).

**Provenance and peer review** Not commissioned; externally peer reviewed.

**Data availability statement** Data are available on reasonable request. Survey data are available from authors on reasonable request. Transcripts of interviews are not available as participants have not given permission to share their data.

**ORCID iDs**
Marta Wanat http://orcid.org/0000-0002-0163-1547

Marta Santillo http://orcid.org/0000-0001-6345-7612
Emily Bongard http://orcid.org/0000-0001-5957-6280
Sinisa Savic http://orcid.org/0000-0001-7910-0554
Christopher C Butler http://orcid.org/0000-0002-0102-3453
Sarah Tonkin-Crine http://orcid.org/0000-0003-4470-1151

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
