## [Reviewer comments · BMJ Open]

ARTICLE DETAILS

TITLE (PROVISIONAL)	Mixed-methods evaluation of a behavioural intervention package to identify and amend incorrect penicillin allergy records in UK general practice
AUTHORS	Wanat, Marta; Santillo, Marta; Galal, Ushma; Davoudianfar, Mina; Bongard, Emily; Savic, Sinisa; Savic, Louise; Porter, Catherine; Fielding, Joanne; Butler, Christopher C.; Pavitt, Sue; Sandoe, Jonathan; Tonkin-Crine, Sarah

VERSION 1 – REVIEW

REVIEWER	Eric Macy Southern California Permanente Medical Group
REVIEW RETURNED	30-Sep-2021

GENERAL COMMENTS	Specific comments: Page 4, line 9: Please consider changing to "...perspective of both patients and clinicians." Page 4, line 28: Please consider changing to "...patient and clinician interview..." Page 4, line 30: Please consider changing to "...perspective of individuals who did..." Page 5, lines 4 to 8: Please consider changing to "...delabeling initiatives. These include providing easier access to skin testing reagents, simplifying and standardizing skin testing protocols for use only in high risk patients, perform routine direct oral challenges in individuals with low-risk histories to confirm current tolerance, and develop a greater appreciation in patients and clinicians of the well-established epidemiology of suspected penicillin allergy, particularly how rarely it is confirmed." Page 5, line 57: Please consider changing to "... (Stage 3), or proceeding directly to an OCT in the vast majority, depending on the..." Page 18, line 12: Please consider changing to "...patient and clinician interview..."
--

REVIEWER	Misha Devchand Austin Health
REVIEW RETURNED	18-Feb-2022

GENERAL COMMENTS	This is an interesting paper that conducted a mixed-methods process evaluation to investigate patients' and clinicians' views on the Penicillin Allergy Testing process. I thought this paper was well written, clear and easy to read. Some questions for the authors to address:
--

	1) Page 8 - was the one clinician from each practice randomly selected? Or was there selection criteria? 2) On Page 8 - under the clinician section, it says "Patients were invited to take part in an interview by letter or telephone. We sampled patients..." please review. 3) On Page 8 - was there a specific number of patients who were invited to take part in the interview? 4) Page 9 - I haven't worked in the UK medical system but could you give some clarity about how the pharmacist was involved in this programme? Are they located in the general practice? Did they also receive the clinician's booklet? 5) Page 9 - "Patients were already confident about the safety and accuracy of the test at baseline, however they were unsure whether they were severely allergic to penicillin and but did not fear a severe allergic reaction to penicillin." Please review this line. Overall, I think this article would be of interest to BMJ open readers and has some interesting findings.
--	--

VERSION 1 – AUTHOR RESPONSE

1	Page 4, line 9: Please consider changing to "...perspective of both patients and clinicians."	We have amended the sentence (page 4, line 13-14)
1	Page 4, line 28: Please consider changing to "...patient and clinician interview..."	We have amended this sentence (page 3, lines 12).
1	Page 4, line 30: Please consider changing to "...perspective of individuals who did..."	We have amended this sentence (page 3, line 13)
1	Page 5, lines 4 to 8: Please consider changing to "...delabeling initiatives. These include providing easier access to skin testing reagents, simplifying and standardizing skin testing protocols for use only in high risk patients, perform routine direct oral challenges in individuals with low-risk histories to confirm current tolerance, and develop a greater appreciation in patients and clinicians of the well-established epidemiology of suspected penicillin allergy, particularly how rarely it is confirmed."	We have amended this sentence and added the suggested text (page 4, lines 5-10).
1	Page 5, line 57: Please consider changing to "... (Stage 3), or proceeding directly to an OCT in the vast majority, depending on the..."	We have amended this sentence (page 5, line 3-4)
1	Page 18, line 12: Please consider changing to "...patient and clinician interview..."	We have amended this sentence (page 17, line 10)
2	This is an interesting paper that conducted a mixed-methods process evaluation to investigate patients' and clinicians' views on the Penicillin Allergy Testing process. I thought this paper was well written, clear and easy to read.	Thank you for your positive feedback
2	1) Page 8 - was the one clinician from each practice randomly selected? Or was there selection criteria?	We have now clarified that one clinician who was invited from each practice was a study

		“champion”, who led on the patient recruitment and was often responsible for updating patients’ records (page 7, lines 10-12).
2	On Page 8 - was there a specific number of patients who were invited to take part in the interview?	We wanted to clarify that we invited 12 patients for an interview. Two have not responded to our invitation (page 12, line 1-2)
2	Page 9 - I haven't worked in the UK medical system but could you give some clarity about how the pharmacist was involved in this programme? Are they located in the general practice? Did they also receive the clinician's booklet?	We wanted to clarify that pharmacists often work in the community but increasingly they are also working as part of the GP practices. This pharmacist has worked in the GP practice which took part in the study and thus was invited to take part.
2	"Patients were already confident about the safety and accuracy of the test at baseline, however they were unsure whether they were severely allergic to penicillin and but did not fear a severe allergic reaction to penicillin. " Please review this line.	This has been corrected (page 10, line 2)
2	Overall, I think this article would be of interest to BMJ Open readers and has some interesting findings.	Thank you for your positive feedback.